# *C. elegans* food choice exhibits effort discounting-like behavior

**Jonathan RM Millet, Serge Faumont, Aaron B Schatz, Amanda M White, Kathy D Chicas-Cruz, Shawn R Lockery***

Institute of Neuroscience, University of Oregon, Eugene, United States

## eLife Assessment

This **important** work develops the *C. elegans* as a model organism for studying effort-based discounting by asking the worms to choose between patches of easy and hard to digest bacteria. The authors provide **convincing** evidence that the nematodes are effort discounting. They also provide **solid** evidence of involvement of dopamine in the food preference and that the finding is not restricted to lab-acclimated strains.

**\*For correspondence:**
shawn@uoregon.edu

**Abstract** Cost–benefit decisions are ubiquitous in both human and animal behavior. Economists have developed formal models of cost–benefit decision-making by focusing on discounting behavior, the devaluation of a reward based on the costs associated with it. The phylogenetic limits of discounting behavior remain unknown. Here, we provide evidence that the nematode *C. elegans* exhibits behavior closely resembling effort discounting. Given a choice between food options that are easy or difficult to consume, worms devalue the latter in a manner predicted by economic models. We identified a plausible mechanism for this behavior based on differential rates of leaving food patches and demonstrated that this mechanism is disrupted by deficits in dopamine signaling, as in rodents. Together, these results establish *C. elegans* as a potential invertebrate model for discounting behavior and set new phylogenetic bounds on this type of cost–benefit decision-making.

## Introduction

Cost–benefit decisions are ubiquitous in animal behavior, driven by universal evolutionary pressures to maximize survival and reproductive success. Foraging, mate selection, parental investment, and territoriality each demand a balance between costs, such as energy expenditure and risk, and benefits such as resources, mates, and offspring. A better understanding of cost–benefit decision-making, including its biological basis, would advance a variety of fields, including economics, psychology, neuroscience, and behavioral ecology.

Economic theorists have developed formal models of cost–benefit decision-making by focusing on *discounting* behavior (**Samuelson, 1937**; **Mazur and Coe, 1987**). In this context, discounting refers to the tendency of individuals to *devalue* a reward based on the costs associated with it. Costs commonly investigated include delay, risk, and effort. In general, as cost increases, perceived or subjective value decreases. For example, in the case of delay discounting, receiving a guaranteed payment of $35 after 30 days is worth less than receiving $25 today if a person accepts the smaller immediate reward. In risk discounting, participants might choose to receive $50 for certain rather than a 50% chance of receiving $100. In effort discounting, participants might choose to receive $1 with no effort rather than $5 after repeatedly squeezing a stiff handgrip (**Stanek and Richter, 2021**).

Studies of discounting are not limited to humans. The literature on discounting in rodents is particularly rich, with multiple paradigms for all three forms of discounting (*Bailey et al., 2016*; *Salamone et al., 2018*; *Gray et al., 2019*; *Castrellon et al., 2021*). In models of delay discounting, for example, mice choose between low- and high-reward nose pokes, where the higher reward arrives after a delay (*Isles et al., 2004*). Risk discounting can be modeled by giving subjects a choice between a low-probability lever yielding a large reward (four food pellets) and a high-probability lever yielding a small reward (one food pellet) 100% of the time (*St Onge et al., 2010*). Effort discounting can be modeled by giving rodents a choice between low-reward (two food pellets) and high-reward (four food pellets) arms of a T-maze where a physical barrier must be climbed to obtain the higher reward (*Salamone et al., 1994*). However, behaviorists have also observed discounting in several species of nonhuman apes, monkeys, lemurs, jays, chickens, pigeons, honeybees, and other animals (*Isles et al., 2004*; *Green et al., 2010*; *Hayden, 2016*).

The phylogenetic limits of discounting behavior are unknown. We therefore asked whether foraging decisions in the nematode worm *C. elegans* exhibit discounting-like behavior. *C. elegans* is an omnivorous bacterivore. Worms swallow bacteria through rhythmic contractions of the pharynx, a tube-shaped muscular organ that delivers bacterial particles to the gut; each pharyngeal contraction is called a pump. *C. elegans* primarily inhabits rotting plant material such as decaying fruits and stems (*Frézal and Félix, 2015*). Its natural habitat contains thousands of different bacterial species (*Samuel et al., 2016*). Each species has a characteristic nutritional quality, defined in terms of the growth rate of individual worms cultured on that species (*Avery and Shtonda, 2003*; *Samuel et al., 2016*). *C. elegans* food preferences are sensitive not only to nutritional quality but also to food density in a manner that satisfies quantitative economic criteria for choices based on subjective value (*Katzen et al., 2023*). *C. elegans* foraging behavior has been used to investigate the neuronal and genetic basis of exploration–exploitation trade-offs (*Bendesky et al., 2011*; *Milward et al., 2011*), a classical example of cost–benefit decision-making. In the laboratory, worms feed on patches of bacteria grown on agarose plates. They leave and re-enter a patch many times. Leaving rates are low on high-quality lawns but increase if the bacterial food is depleted or is difficult to ingest (*Shtonda and Avery, 2006*; *Bendesky et al., 2011*; *Milward et al., 2011*; *Olofsson, 2014*; *Scheer and Bargmann, 2023*). This response pattern suggests that leaving promotes exploration to find a better food source.

The fact that *C. elegans* exhibits subjective-value-based choice and cost–benefit feeding decisions suggested to us that it might also exhibit discounting behavior. Of the various forms of discounting, we focused on effort discounting as it uniquely does not require learned instrumental responses, like lever pressing for rewards, which are not feasible in *C. elegans*. To investigate discounting behavior, we baited the arms of a miniature T-maze with selected densities of a single species of bacteria. One arm contained normal bacteria, while the other arm contained a bacterial preparation in which the cells had been elongated, which we predicted would make them more effortful to ingest and less preferred by *C. elegans*.

We found that *C. elegans* exhibits choice behavior that closely resembles effort discounting. Given a choice between equal densities of normal and elongated bacteria, worms prefer normal bacteria. This preference can be reversed by presenting elongated bacteria at a higher density than normal bacteria, indicating that high food density compensates for greater effort, as in rodent models. Furthermore, worms can be made indifferent to the two food options by adjusting relative food densities. It is therefore possible to measure the extent of devaluation and accurately predict novel indifference points. Preference for normal bacteria can be attributed to an increased patch-leaving rate on elongated food. Additionally, typical levels of discounting require intact dopamine signaling. Our results expand the repertoire of cost–benefit behaviors in *C. elegans* and demonstrate that lower invertebrates are capable of effort discounting-like behavior.

## Results
### Elongated bacteria require more effort to eat

We conceptualized feeding effort in *C. elegans* as the amount of pharyngeal muscle activity required to ingest a given quantity of food. The starting point of our experiments was to make bacteria more effortful to consume. Computer simulations of pharyngeal mechanics predict that large bacterial cells are swallowed less efficiently than small cells (*Avery and Shtonda, 2003*). Consistent with this result,

**Table 1.** Statistical information.

Bold p-values denote significant effects. Note that the Tukey and Dunnett tests are multiple comparison tests and their p-values do not need to be adjusted for multiple comparisons. Replicates were biological unless otherwise indicated.

| Row | Figure | Test | Effect or comparison tested | Units of replication or sampling | Number of replicates | Statistic | Statistic value | DF 1 or combined DF | p | Effect size metric | Effect size |
|---|---|---|---|---|---|---|---|---|---|---|---|
| 1 | 1A | Mann–Whitney | Cephalexin | Worms | 99, 98 | $U$ | 4178 | – | 9.23E−02 | Rank-biserial correlation ($r$) | 0.13 (small) |
| 2 | 1B2 | Mann–Whitney | Cephalexin | Worms | 44, 49 | $U$ | 618 | – | **3.20E−04** | Rank-biserial correlation ($r$) | 0.43 (moderate) |
| 3 | 1C | $t$-Test | Cephalexin | Biol. Replicate | 3, 3 | $t$ | 0.1 | 2 | 9.28E−01 | Cohen's $d$ | 0.08 (small) |
| 4 | 1D | $t$-Test | Cephalexin | Biol. Replicate | 3, 3 | $t$ | 0.1 | 2 | 9.31E−01 | Cohen's $d$ | 0.08 (small) |
| 5 | 1E | $t$-Test | Cephalexin | Tech. Replicate | 3, 3 | $t$ | 0.12 | 2 | 9.91E−01 | Cohen's $d$ | 0.01 (small) |
| 6 | 1F2 | $t$-Test | Mean not = 0 | Assay plates | 108 | $t$ | 1.65 | 104 | 3.56E−01 | Cohen's $d$ | 0.16 (small) |
| 7 | 2B | One-way ANOVA | OD ratios | Assay plates | 137, 106, 73 | $F$ | 19.07 | 2 | **1.52E−08** | Eta squared | 0.11 (moderate) |
| 8 | 2B | $t$-Test | Mean not = 0 | Assay plates | 137 | $t$ | 3.2 | 136 | **1.69E−03** | Cohen's $d$ | 0.27 (small) |
| 9 | 2B | $t$-Test | Mean not = 0 | Assay plates | 73 | $t$ | 1.99 | 72 | **2.06E−07** | Cohen's $d$ | 0.67 (large) |
| 10 | 2B | $t$-Test | Mean not = 0 | Assay plates | 106 | $t$ | 1.08 | 105 | 2.82E−01 | Cohen's $d$ | 0.10 (small) |
| 11 | 2C | One-way ANOVA | OD ratios | Assay plates | 120, 106, 132 | $F$ | 0.83 | 2 | 4.37E−01 | Eta squared | 0.00 (negligible) |
| 12 | 2C | $t$-Test | Mean not = 0 | Assay plates | 120 | $t$ | 0.71 | 119 | 4.80E−01 | Cohen's $d$ | 0.06 (small) |
| 13 | 2C | $t$-Test | Mean not = 0 | Assay plates | 106 | $t$ | 1.08 | 105 | 2.82E−01 | Cohen's $d$ | 0.10 (small) |
| 14 | 2C | $t$-Test | Mean not = 0 | Assay plates | 132 | $t$ | 0.64 | 131 | 5.22E−01 | Cohen's $d$ | 0.06 (small) |
| 15 | 3C | $t$-Test | $P_F$ vs ($P_O$, $P_B$) | Single worm | 66, 60 | $t$ | 4.22 | 123.57 | **4.62E−05** | Cohen's $d$ | 0.75 (medium) |
| 16 | 3C | $t$-Test | $P_O$ vs $P_B$ | Single worm | 66, 60 | $t$ | 0.57 | 116.99 | 5.73E−01 | Cohen's $d$ | 0.10 (small) |
| 17 | 3D | One-way ANOVA | Transition type | Single worm | 66, 60 | $F$ | 24.71 | 3 | **4.22E−14** | Eta squared | 0.22 (large) |
| 18 | 3D | Tukey | $k_{FB}$, $k_{BF}$ | Single worm | 66, 60 | $q$ | 3.68 | 122 | **<1.00E−07** | Cohen's $d$ | 2.199391 (large) |
| 19 | 3D | Tukey | $k_{BO}$, $k_{OB}$ | Single worm | 66, 60 | $q$ | 3.68 | 122 | **4.30E−06** | Cohen's $d$ | 0.68 (medium) |
| 20 | 3D | Tukey | $k_{BO}$, $k_{BF}$ | Single worm | 66, 60 | $q$ | 3.68 | 122 | **6.00E−07** | Cohen's $d$ | 1.60 (large) |
| 21 | 3D | Tukey | $k_{OB}$, $k_{BF}$ | Single worm | 66, 60 | $q$ | 3.68 | 122 | 9.77E−01 | Cohen's $d$ | 0.051 (negligible) |
| 22 | 3D | Tukey | $k_{FB}$, $k_{BO}$ | Single worm | 66, 60 | $q$ | 3.68 | 122 | 4.86E−01 | Cohen's $d$ | 0.91 (large) |
| 23 | 3D | Tukey | $k_{OB}$, $k_{FB}$ | Single worm | 66, 60 | $q$ | 3.68 | 122 | **0** | Cohen's $d$ | 0.88 (large) |
| 24 | 3D | MANOVA | Cephalexin | Single worm | 66, 60 | $F$ | 16.44 | 4121 | **8.77E−11** | Eta squared (partial) | 0.35 (large) |
| 25 | 3D1 | $t$-Test | Cephalexin | Single worm | 66, 60 | $t$ | 7.38 | 79.39 | **1.36E−10** | Cohen's $d$ | 1.36 (large) |
| 26 | 3D2 | $t$-Test | Cephalexin | Single worm | 66, 60 | $t$ | 3.32 | 118.69 | **1.18E−03** | Cohen's $d$ | 0.60 (medium) |
| 27 | 3D3 | $t$-Test | Cephalexin | Single worm | 66, 60 | $t$ | 0.28 | 123.25 | 7.83E−01 | Cohen's $d$ | 0.05 (negligible) |
| 28 | 3D4 | $t$-Test | Cephalexin | Single worm | 66, 60 | $t$ | 1.38 | 95.7 | 1.70E−01 | Cohen's $d$ | 0.24 (small) |
| 29 | 3E | Two-way ANOVA | Cephalexin | Single worm | 66, 60 | $F$ | 5.24 | 1 | **2.29E−02** | Eta squared | 0.02 (small) |
| 30 | 3E | Two-way ANOVA | Food | Single worm | 66, 60 | $F$ | 0.7 | 1 | 4.00E−01 | Eta squared | 0.00 (negligible) |
| 31 | 3E | Two-way ANOVA | Cephalexin × Food | Single worm | 66, 60 | $F$ | 6.66 | 1 | **1.10E−02** | Eta squared | 0.03 (small) |
| 32 | 3E | $t$-Test | Cephalexin | Single worm | 66, 60 | $t$ | 3.68 | 115.91 | **4.59E−04** | Cohen's $d$ | 0.66 (medium) |

*Table 1 continued on next page*

*Table 1 continued*

| Row | Figure | Test | Effect or comparison tested | Units of replication or sampling | Number of replicates | Statistic | Statistic value | DF 1 or combined DF | p | Effect size metric | Effect size |
|---|---|---|---|---|---|---|---|---|---|---|---|
| 33 | 3E | *t*-Test | Cephalexin | Single worm | 66, 60 | *t* | 0.19 | 123.94 | 8.47E−01 | Cohen's *d* | 0.034 (negligible) |
| 34 | 4 | One-way ANOVA | Strain | Assay plates | 137, 179, 165, 160, 175, 180 | *F* | 7.69 | 3 | **4.71E−05** | Eta squared | 0.03 (small) |
| 35 | 4 | Dunnett test | | | | *q* | 2.5 | 649 | | | |
| 36 | | | N2 vs cat-2 | Assay plates | 137, 179 | *q* | 0.97 | | 7.72E−01 | | |
| 37 | | | N2 vs dat-1 | Assay plates | 137, 164 | *q* | 1.3 | | 5.37E−01 | | |
| 38 | | | N2 vs dop-1 | Assay plates | 137, 180 | *q* | −1.61 | | 3.33E−01 | | |
| 39 | | | N2 vs dop-2 | Assay plates | 137, 176 | *q* | 1.25 | | 5.71E−01 | | |
| 40 | | | N2 vs dop-3 | Assay plates | 137, 160 | *q* | 2.78 | | **1.92E−02** | Cohen's *d* | 0.50 (medium) |
| 41 | 4 | *t*-Test | Mean not = 0 | Assay plates | 180 | *t* | 0.65 | 159 | 5.14E−01 | Cohen's *d* | 0.05 (small) |
| 42 | 5 | One-way ANOVA | Strain | Assay plates | 137, 137, 129, 184, 178, 146 | *F* | 3.3 | 5 | **5.83E−03** | Eta squared | 0.018 (small) |
| 43 | 5 | Dunnett test | | | | *q* | 2.51 | 905 | | | |
| 44 | | | N2 vs DL238 | Assay plates | 137, 137 | *q* | 0.31 | | 9.98E−01 | | |
| 45 | | | N2 vs CB4856 | Assay plates | 137, 129 | *q* | 0.51 | | 9.81E−01 | | |
| 46 | | | N2 vs JU258 | Assay plates | 137,184 | *q* | 2.26 | | 9.42E−02 | | |
| 47 | | | N2 vs MY23 | Assay plates | 137, 178 | *q* | 2.46 | | 5.81E−02 | | |
| 48 | | | N2 vs JU775 | Assay plates | 137, 146 | *q* | 3.03 | | **1.12E−02** | Cohen's *d* | 0.68 (medium) |
| 49 | 5 | *t*-Test | Mean not = 0 | Assay plates | 146 | *t* | 0.36 | 145 | 1.03E−01 | Cohen's *d* | 0.13 (small) |
| 50 | 5 | *t*-Test | Mean not = 0 | Assay plates | 137 | *t* | 2.02 | 136 | **4.56E−02** | Cohen's *d* | 0.17 (small) |

worms develop more slowly when cultured on large bacteria, possibly because they are acquiring nutrients less efficiently (*Avery and Shtonda, 2003*). These findings led to the hypothesis that larger bacteria require more effort to eat, necessitating more pharyngeal pumps per unit of food consumed. To test this hypothesis, we artificially increased the size of an easily ingested bacterium, *Comamonas* spp., using the antibiotic cephalexin. In Gram-negative bacteria such as *Comamonas*, cephalexin interferes with bacterial cell septation during mitosis, resulting in elongated bacterial filaments that nevertheless remain edible (*Millet et al., 2022*; *Figure 1—figure supplement 1*).

Pharyngeal pumping rates in normal and elongated bacteria were measured in the reference strain N2 by placing individual worms in a microfluidic channel filled with a bacterial suspension and fitted with electrodes that record the electrical signals associated with each pharyngeal contraction (*Lockery et al., 2012*). Bacteria were washed with cephalexin-free buffer before being fed to worms. Pumping frequencies in normal and elongated bacteria were equivalent (*Figure 1A*; *Table 1*, row 1), indicating that worms expend approximately the same amount of effort per unit time feeding on normal and elongated food. To test the hypothesis that elongation interferes with food absorption, we measured nutrient uptake in worms cultured on normal or elongated bacteria, using fat stores as a proxy for carbohydrate and lipid consumption. This was done by staining fat stores with the specific label Oil-Red-O (*Soukas et al., 2009*) and quantifying the density of staining after a 2-day culture period extending from the L1 to the L4 stage. We found that fat levels were significantly lower in worms cultured on elongated food (*Figure 1B2*; *Table 1*, row 2). Given that the pumping rate was unchanged in the presence of normal versus elongated bacteria (*Figure 1A*), the lower fat stores observed in worms fed elongated bacteria suggest that the amount of nutrition incorporated per pump is lower in worms feeding on elongated versus normal bacteria. This supports our hypothesis that elongation impairs food transport, requiring longer feeding bouts and thus increased muscular activity to consume an equivalent amount of food.



**Figure 1.** Establishment of an effort-discounting paradigm in *C. elegans*. (**A**) Pharyngeal pumping frequency in suspensions of normal (CON) and elongated (CEPH) bacteria. Symbols (box and whisker plot): *filled circles*, single-worm data points; *diamonds*, means; *numbers*, sample size. (**B1**) Quantification of Oil-Red-O staining of lipid in worms cultured on normal (CON) or elongated (CEPH) bacteria. Median worms from each condition are shown. (**B2**) Mean pixel intensity of Oil-Red-O staining of lipid in worms cultured on normal (CON) or elongated (CEPH) bacteria. *Diamonds*, means;

*Figure 1 continued on next page*

*Figure 1 continued*

*numbers*, sample size; *, *p* = 3.20E-04. (**C, D**) Protein and carbohydrate in worms cultured on normal (CON) and elongated bacteria (CEPH). Columns of data points of the same color are biological replicates run in parallel. Symbols: *filled circles*, technical replicates (*n* = 6); *diamonds*, means of technical replicates. (**E**) Lipid concentration in worms cultured on normal (CON) and elongated bacteria (CEPH). Symbols: *filled circles*, biological replicates (*n* = 3); *diamonds*, means of biological replicates. (**F1**) T-maze assay for relative attractiveness of normal and elongated bacteria. Food patches contained sodium azide as a paralytic agent. (**F2**) Mean preference index in the T-maze assay. Error bars, ± 95% CI.

The online version of this article includes the following figure supplement(s) for figure 1:

**Figure supplement 1.** Cephalexin-treated bacteria.

Our effort-discounting paradigm assumes that the normal and elongated bacteria are equivalent except for the amount of effort required to consume them. For example, elongation could affect nutrient content or olfactory attractiveness of bacteria. *C. elegans* food preferences are sensitive to the nutrient content of bacteria (*Feng et al., 2025*). We therefore asked whether cephalexin treatment affects nutrient content. We focused on key macronutrients: carbohydrates, lipids, and proteins. For protein and carbohydrates, we used colorimetric assays (Materials and methods; *Figure 1C, D*). For lipids, we used phase separation to harvest and measure lipid content (Materials and methods; *Figure 1E*). In all three tests, we detected no differences between control and cephalexin-treated bacteria (*Table 1*, rows 3–5). We conclude that cephalexin treatment does not alter bacterial nutritional composition.

Worms are attracted to particular species of bacteria by the characteristic mixture of volatile organic compounds each species emits (*Worthy et al., 2018*). Thus, another concern is that cephalexin treatment might alter the odors produced by bacteria, thereby changing their attractiveness. To control for this possibility, we performed a T-maze assay (Materials and methods) in which patches of normal and elongated bacteria were placed at the ends of each respective arm. The patches contained equal volumes of bacteria at an optical density (OD) of 1.0. Patches were spiked with sodium azide, a paralytic agent, to trap worms once they reached a patch. Worms were placed at the base of the T-maze and allowed to explore it for 60 min after which the number of worms in each patch was counted. A food preference index ($-1 \leq I \leq 1$) was calculated, where positive and negative values indicate preference for normal and elongated bacteria, respectively, and $I \cong 0$ denotes indifference between the two options (Materials and Methods). We found that worms under these conditions were indifferent (*Figure 1F2*; *Table 1*, row 6). This result indicates that cephalexin treatment does not alter the relative attractiveness of normal or elongated bacteria.

## Worms devalue food that requires more effort to eat

Having demonstrated that elongated bacteria require more effort to eat but are equivalent to normal bacteria in terms of nutrition and attractiveness, we next asked whether *C. elegans* exhibits a preference for the normal, easier-to-eat bacteria – evidence of effort discounting-like behavior. We tested this by measuring food preference in the T-maze assay using the same methodology as in *Figure 1F* but without the paralytic agent, allowing individual worms to sample both patches. One arm of the T-maze was baited with normal bacteria at an OD of 1.0, while the other arm contained elongated bacteria at ODs of 1.0, 1.5, or 2.0. Worms explored the maze for 60 min and the food preference index was computed based on worm counts at the end of this period. There was a significant overall effect of OD of elongated bacteria on preference (*Figure 2B*; *Table 1*, row 7). When normal and elongated bacteria were at the same concentration (OD 1), worms preferred normal bacteria (*I* = 0.10; *Figure 2B*; *Table 1*, row 8). This demonstrates that worms do, in fact, devalue elongated food. When normal and elongated bacteria were at OD 1.0 and 2.0, respectively, worms preferred elongated bacteria (*I* = –0.17; *Figure 2B*; *Table 1*, row 9). This suggests that, in the worm's valuation system, higher food density compensates for greater effort. This outcome was not a foregone conclusion. For instance, it was possible that no amount of elongated food would be sufficient to make worms prefer it. Our results are consistent with findings in rodents, where high-effort options are chosen when the food reward density is sufficiently high (*Floresco et al., 2008*). When the optical densities of normal and elongated food were 1.0 and 1.5, respectively, worms were indifferent between the two options (*I* = 0.03; *Figure 2B*; *Table 1*, row 10). Indifference points are also observed in rodent studies of effort discounting (*Salamone et al., 1994*; *van den Bos et al., 2006*; *Ostrander et al., 2011*). Overall, we

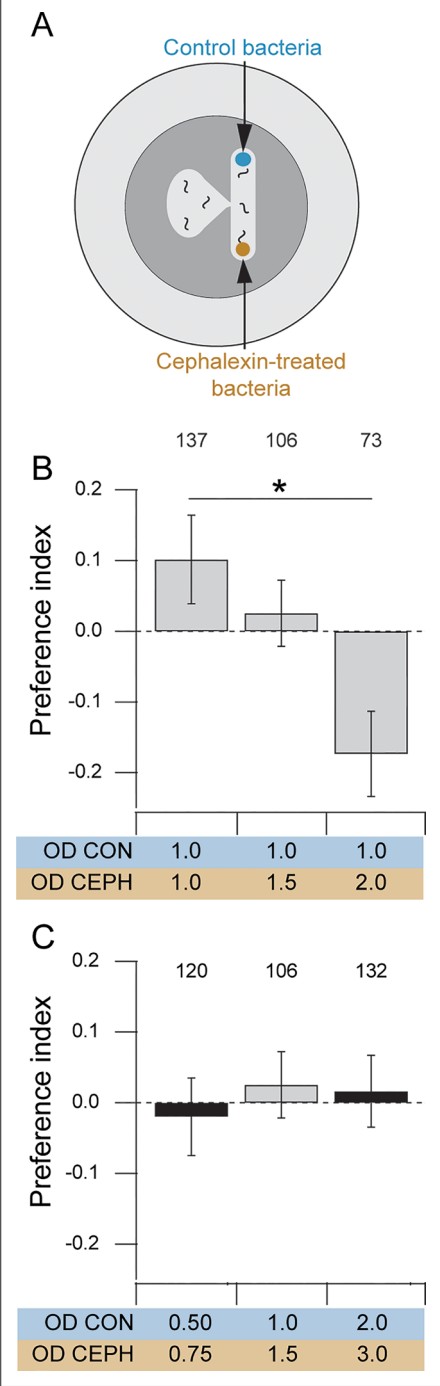

**Figure 2.** Food preference in T-maze assays. (**A**) T-maze assay for preference for normal (Control) versus elongated bacteria (cephalexin). (**B**) Effect on preference of raising the optical density of elongated

*Figure 2 continued*

bacteria (CEPH). Mean preference index is shown for assays in which three different optical densities (OD) of elongated bacteria (CEPH) were paired with normal bacteria (CON) at optical density (OD) 1.0. Worms are indifferent when elongated bacteria are at OD 1.5. *Numbers*, sample size; *, $p = 1.52E-08$. Error bars ± 95% CI. (**C**) Tests of predicted indifference points. Mean preference index is shown for three different pairs of optical densities of normal (CON) and elongated (CEPH) bacteria. The gray bar is the indifference point found in panel B. The black bars are novel indifference points found by scaling the optical density of elongated bacteria by the discount factor of 2/3 obtained from the gray bar. *Numbers*, sample size. Error bars ± 95% CI.

The online version of this article includes the following figure supplement(s) for figure 2:

**Figure supplement 1.** Standard models of effort discounting.

conclude that in feeding decisions, worms take into account both relative effort and relative food density.

## An economic model predicts novel indifference points

Indifference points are significant because, by definition, the presented options have equal perceived value, allowing for the quantification of devaluation. This quantification is only possible at an indifference point. For example, consider a hypothetical human scenario in which a participant is asked to choose between receiving $20 immediately or $20 with the added cost of climbing 20 flights of stairs. Most participants would choose the immediate $20. Given a choice between $20 immediately and $100 at the cost of climbing 20 flights of stairs, some participants might choose the $100 option. However, suppose that when given a choice between $20 immediately and $80 at the cost of climbing 20 flights of stairs, a particular participant is indifferent between the two options. For this person, under these circumstances, we can conclude that the $20 option is perceived as equal in value to the $80 option. In other words, the $80 option has been discounted (devalued) to $20.

Formally, at an indifference point,

$$V(E) = d(E) V_0$$

where $E$ represents effort, $V_0$ represents the original value of the reward before devaluation, and $V(E)$ is the discounted value of $V_0$. The quantity $d(E)$ is the discount factor, which, in economic models of discounting, is a function of $E$. From the above equation,

$$d(E) = \frac{V(E)}{V_0}$$

with $0 \leq d(E) \leq 1$; in the above example $d(E)$ = 20/80 or 1/4. In economic models, $d(E)$ is inversely related to $E$ but the precise form of $d(E)$ varies between models (**Figure 2—figure supplement 1**).

In our experiments, $E$ is determined by the degree of bacterial elongation. Although this quantity is unknown, it remained unchanged because we used the same cephalexin exposure protocol across experiments (see Materials and methods). It is analogous to the fixed effort of climbing 20 flights of stairs in the hypothetical example. Assuming a linear relationship between OD and perceived value, we can determine $d(E)$ from the indifference point shown in **Figure 2B**, where $V_0$ = OD 1.5 and $V(E)$ = OD 1.0, yielding $d(E)$ = 2/3. In the most basic models of discounting, $d(E)$ is independent of $V_0$. Assuming independence, other values of $V_0$ can be inserted into equation 1 to predict novel indifference points. For instance, setting $V_0$ = OD 3 predicts indifference at $V(E)$ = OD 2, and setting $V_0$ = OD 0.75 predicts indifference at $V(E)$ = OD 0.5. That is, any pair of optical densities (normal versus elongated) in a 2:3 ratio should be an indifference point. We tested these predictions by measuring preference in T-mazes baited with normal and elongated food at OD ratios (normal:elongated) of 2:3 and 0.5:0.75, finding indifference as predicted (**Figure 2C**, **Table 1**, rows 11–14). The fact that standard discounting models correctly predict indifference points reinforces the conclusion that *C. elegans* exhibits discounting-like behavior.

## Effort discounting-like behavior is based on assessment of the local food environment

We next investigated the behavioral mechanism through which effort influences food preference, examining how worms interact with patches of normal versus elongated food. We studied these interactions at the microscopic level of individual worms to explain behavior at the macroscopic level of T-maze assays. This was done by performing food-patch-leaving assays (**Scheer and Bargmann, 2023**) using either normal or elongated bacteria. Individual worms were transferred to a 1-cm diameter, nematode growth medium (NGM)-filled arena containing a small patch (3–4 mm diameter) of normal or elongated bacteria (OD 1.0) at its center. The patch size was comparable to those in the T-maze assay. Each worm was video recorded for 15 min while it entered and left the patch multiple times (**Figure 3A**).

For analysis, we took a kinetic approach to investigate the rates at which worms stochastically changed their position relative to the food patch. This approach is both simple, as it does not require assigning specific locomotory states at particular times to individual worms, and powerful, as it forms a bridge between the microscopic and macroscopic behavior of a stochastic system, thereby elucidating underlying dynamics. Accordingly, we defined three kinetic states based on occupancy of specific zones in the arena (**Figure 3B**):

1. State F, *On Food*: both head and tail of the worm were in contact with the food patch.
2. State B, *At Border*: either the head or tail was in contact with the food patch, but not both.
3. State O, *Off Food*: neither head nor tail was in contact with the food patch.

This system contains four state transitions: F → B, B → O, O → B, and B → F. Behavior was analyzed in terms of state probabilities ($P_F$, $P_B$, $P_O$) and rate constants ($k_{FB}$, $k_{BO}$, $k_{OB}$, $k_{BF}$). State probability was defined as the fraction of total observation time the worm was in each state. Rate constants were defined as the number of transitions divided by the total time spent in the state of origin. Rate constants determine the probability per unit time of transitioning from one state to another and thus reflect the underlying physical causes of state transitions.

We observed a variety of changes in kinetic parameters on elongated versus normal food. $P_B$ and $P_O$ increased at the expense of $P_F$, consistent with decreased preference for elongated food in T-maze

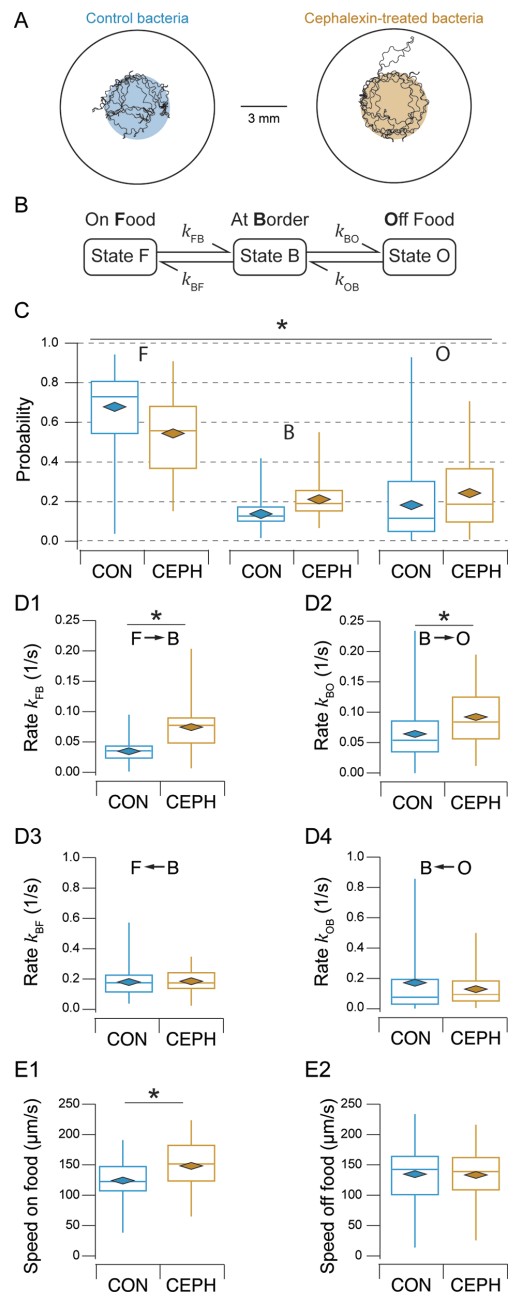

**Figure 3.** Kinetic analysis of food-patch-leaving assays. (**A**) Food-patch-leaving assays for normal (control) and elongated (cephalexin-treated) bacteria. *Open circles*, arenas; *colored circles*, food. A representative track of a single worm is superimposed on each diagram. (**B**) Three-state kinetic model of behavior in the assay. Each state represents the indicated zone in the arena. (**C**) Probability of states F, B, and O for normal (CON) and elongated (CEPH) bacteria. Significance was assessed using compositional statistics as described in *Figure 3—figure supplement 1*. *, p = 4.62-E-05. (**D**) Rate constants for normal (CON) and elongated (CEPH) bacteria. *, *p* < 1.19E-03. (**E**) Locomotion speed

*Figure 3 continued on next page*

*Figure 3 continued*

for normal (CON) and elongated (CEPH) bacteria on or off food. *, *p* = 4.59E-04.

The online version of this article includes the following figure supplement(s) for figure 3:

**Figure supplement 1.** Statistical test of the effect of cephalexin on state probabilities in *Figure 3C*.

assays (*Figure 3C*; *Table 1*, rows 15 and 16). The rate constants exhibited two key features. First, in control animals, between adjacent states, the mean rate constants leading to patch entry (O → B → F transitions) were greater than the mean rate constants leading to patch leaving (F → B → O transitions). In particular, $k_{BF} > k_{FB}$ and $k_{OB} > k_{BO}$ (*Figure 3, D1 vs D3 and D2 vs D4*; *Table 1*, rows 17–23). This indicates a greater propensity to enter food patches than to leave them, which is consistent with efficient foraging. Second, changes in rate constants brought about by elongated food exhibited a distinctive pattern: the rates for patch leaving ($k_{FB}, k_{BO}$) were increased whereas the rates for patch entry ($k_{OB}, k_{BF}$) were unchanged (*Figure 3, D1 and D2 vs D3 and D4*; *Table 1*, rows 24–28). Thus, worms exhibited a heightened propensity to leave elongated food, consistent with previous observations (*Scheer and Bargmann, 2023*). The fact that rates for patch entry were unaffected by elongated bacteria rules out a mechanism in which worms, exiting patches of elongated food (which is relatively undesirable), become reluctant to re-enter them. Finally, we found that speed on elongated food increased, whereas speed off food remained unchanged (*Figure 3E*; *Table 1*, rows 29–33). Applying the kinetic data of individual worms to the population level in T-maze assays suggests that effort discounting-like behavior could result from feeding decisions based on the local food environment rather than a direct comparison of patch contents across the arms of the T-maze.

## Intact dopamine signaling is required for effort discounting-like behavior

A salient aspect of effort discounting in rodents is its sensitivity to changes in midbrain dopamine signaling (*Salamone et al., 2018*). Increased signaling promotes preference for high-value, high-effort rewards, whereas decreased signaling promotes preference for low-value, low-effort rewards (*Bailey et al., 2016*; *Salamone et al., 2018*; *Hart and Izquierdo, 2019*); in other words, increased dopamine signaling results in

greater effort tolerance. Dopamine signaling is a candidate mechanism for the regulation of effort discounting-like behavior in *C. elegans* because it affects how worms respond to the presence and absence of food as they enter or leave food patches. Worms slow down when they encounter a food patch, a response that requires intact dopamine signaling (*Sawin et al., 2000*). When worms exhaust their food supply or are removed from a food patch, they engage in a bout of frequent, high-angled turns called area-restricted search (*Hills et al., 2004*). This response also requires dopamine signaling.

To test whether dopamine signaling is involved in effort discounting-like behavior in *C. elegans*, we examined food preference in a variety of mutants with altered dopamine signaling. The gene *cat-2* encodes a tyrosine hydroxylase essential to the main synthesis pathway for dopamine; *cat-2* mutants are commonly used to reduce dopamine signaling in *C. elegans*. The gene *dat-1* encodes a dopamine transporter that mediates the reuptake of dopamine into presynaptic neurons; *dat-1* mutants are used to increase dopamine signaling. The effects of reduced dopamine signaling can also be investigated in dopamine receptor mutants. *C. elegans* has four genes encoding dopamine receptors with homology to dopamine receptors in mammals. Of these, *dop-1*, *dop-2*, and *dop-3* are the most thoroughly characterized to date. *dop-1* is a DRD1 homolog, whereas *dop-2* and *dop-3* are DRD2 homologs. The genes *dop-1* and *dop-3* are required for normal responses to the presence of food (*Chase et al., 2004*); *dop-2* is required for increased levels of activity in the absence of food (*Bastien et al., 2024*). We found a significant overall effect of strain on food preference measured in the T-maze assay when both foods were at OD 1.0 (*Figure 4*; *Table 1*, row 34). Post hoc testing showed that *cat-2*, *dat-1*, *dop-1*, and *dop-2* were indistinguishable from N2, whereas *dop-3* mutants were significantly different from N2 (*Table 1*, rows 36–40). In particular, they were indifferent between the two food options, indicative of decreased effort discounting-like behavior (*Table 1*, row 41). We conclude that fully intact dopamine signaling is required for normal effort discounting-like behavior in *C. elegans*, as in rodents. However, the fact that impaired dopamine signaling rendered worms indifferent between the food options suggests that decreased dopamine signaling increases tolerance for high-effort food, in contrast to its effect in rodents. The absence of an effect of the *cat-2* and *dat-1* mutations is puzzling in view of the requirement for *dop-3*. One possibility is that residual dopamine in the *cat-2* mutant (*Sanyal et al., 2004*) is sufficient for normal effort discounting-like behavior and that normal levels of dopamine signaling are saturating, such that *dat-1* mutations have no effect.

## Effort discounting-like behavior may not be an adaptation to the laboratory environment

Finally, we considered whether effort discounting-like behavior may reflect adaptation to the laboratory environment. The reference strain used in this study, N2, has been cultured for over 50 years in the laboratory setting, where it is typically fed small, easily consumed bacteria, such as *Escherichia coli* strain OP50. Consequently, it may have lost the ability to consume larger bacteria efficiently. To address this question, we tested whether any natural isolate strains would prefer normal over elongated bacteria. Accordingly, we performed T-maze assays on five such strains with normal and elongated food at OD 1.0. Natural isolate strains proved to be challenging to work with as they frequently escaped the maze or burrowed in the substrate. These traits reduced the number of countable worms per assay, leading to considerable variance in the data. Nevertheless, there was a significant overall effect of strain on mean preference index (*Figure 5*; *Table 1*, row 42). Post hoc testing revealed that only the strain JU775 had a preference index that was significantly different from N2 (*Table 1*, row 48) and that JU775's preference was not distinguishable from zero (*Table 1*, row 49). Thus, JU775 was indifferent between normal and elongated food. On the other hand, mean preference of DL238 was both different from zero (*Table 1*, row 50) and indistinguishable from N2 (*Table 1*, row 44). That is, DL238 behaved like N2. We conclude that N2's preference is probably not an adaptation to the laboratory environment. Between DL238 and JU775, there was a range of different responses, suggesting the possibility of genetic variation in preference for normal versus elongated food. This could reflect differences in effort tolerance or ease of consumption of elongated bacteria.

## Discussion

Effort discounting is a form of cost–benefit decision-making with implications for economics, psychology, neuroscience, behavioral ecology, and other fields. To investigate effort discounting in *C.*

*elegans*, worms were given a choice between two patches of the same species of bacteria. One patch contained normal bacteria, whereas the other patch contained elongated bacteria, which we showed to be more effortful to consume yet similar to normal bacteria in nutrition and attractiveness. When the two types of bacteria were presented at equal density, worms preferred normal bacteria. Three lines of evidence suggest that this preference is analogous to the devaluation of a reward based on increased effort to obtain it: (1) Preference can be reversed by increasing the density of the effortful option. (2) The relative density of normal and effortful food can be titrated to an indifference point. (3) Widely used economic models of discounting correctly predict novel indifference points in *C. elegans*. Points (1) (*Floresco et al., 2008*) and (2) (*Salamone et al., 1994*; *van den Bos et al., 2006*; *Ostrander et al., 2011*), together with the requirement for intact dopamine signaling (*Salamone et al., 2018*), establish similarities to effort discounting in other animal models. The demonstration of effort discounting-like behavior in a lower invertebrate sets a new phylogenetic boundary on discounting behavior.

### Relationship to previous work

An increase in patch-leaving frequency in response to elongated bacteria has been reported previously (*Scheer and Bargmann, 2023*). There are some significant differences between that study and ours: (1) *E. coli* was used instead of *Comamonas*; (2) bacteria were elongated with the antibiotic aztreonam which renders food inedible to *C. elegans* (*Gruninger et al., 2008*; *Ben Arous et al., 2009*), whereas the cephalexin treatment we used resulted in elongated but edible food (*Millet et al., 2022*); (3) worms were video recorded for 40 min instead of 15 min; (4) food-patch leaving was defined using different behavioral criteria; (5) patch-entry rates were not considered. Nevertheless, our results confirm and extend the previous findings by showing that the effect of elongated food on patch-leaving rate is robust across bacterial species, varied degrees of edibility of elongated food, and differences in details of behavioral analysis.

### Utility of economic modeling

Although *C. elegans* patch-leaving behavior has been described in detail before, placing it in an explicit economic context is new. This context provided a quantitative framework that enriched our study in two respects. First, it gave us a metric – the discount factor – for quantifying the extent to which effortful food was devalued by the worm. We observed that worms were indifferent when normal and elongated bacteria were at ODs 1.0 and 1.5, respectively, yielding a discount factor of 2/3.

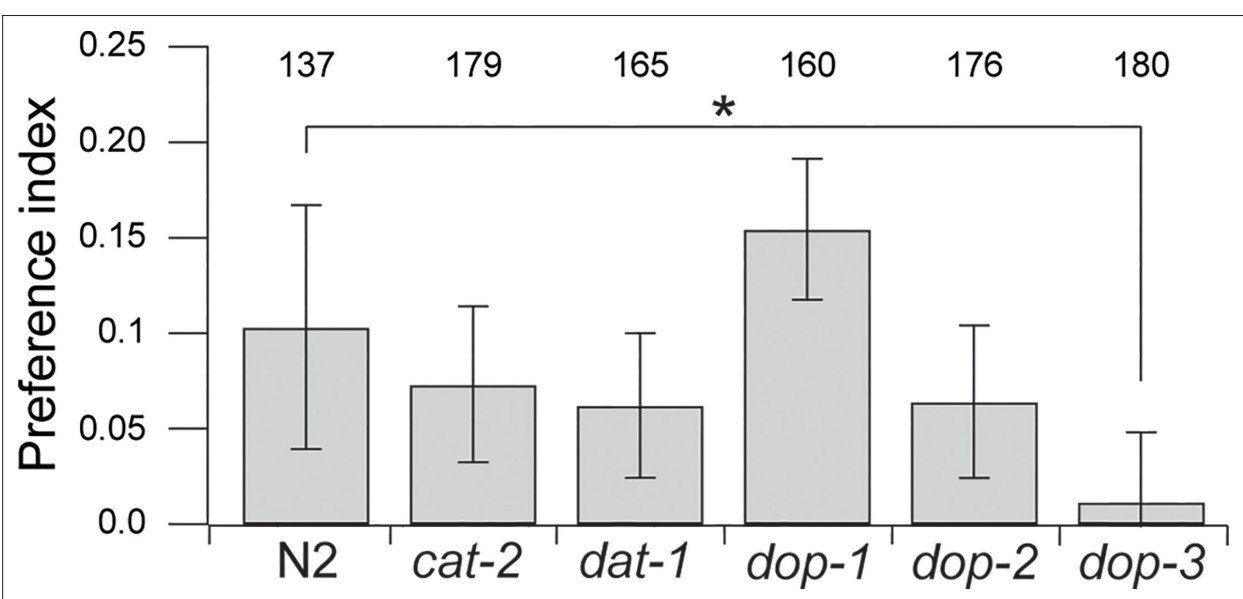

**Figure 4.** Effect of dopamine signaling mutations on preference for normal versus elongated bacteria. Data are mean preferences measured in T-maze assays. Dopamine signaling was reduced by a nonsense mutation in *cat-2* or by deletions of three dopamine receptor genes (*dop*). Dopamine signaling was increased by a deletion in *dat-1*. *Numbers*, sample size; *, $p$ = 1.92E-02. Error bars ± 95% CI.

Second, this framework made testable predictions. At the level of effort determined by our cephalexin exposure protocol, any combination of optical densities of normal and elongated food in a 2:3 ratio should be an indifference point. This prediction was verified by measuring preference under conditions in which the OD of elongated food was both higher and lower than the original value of 1.5. This result shows that standard models of effort discounting may apply to *C. elegans*. It will now be interesting to test whether the discount factor is sensitive to effort, as assumed in many discounting models (*Figure 1—figure supplement 1*). This could be done by identifying indifference points when effort is manipulated by changing cephalexin exposure conditions to increase or decrease the extent of elongation.

## A model of effort discounting-like behavior

We propose a model of effort discounting that builds upon previous studies of foraging in *C. elegans*. Once worms left the starting point of the T-maze, they were free to move back and forth between food patches at the end of each arm, encountering one patch then the other in serial fashion. Thus, our procedure was a foraging assay. In a simple model to explain the food preferences we observed, worms make an on-line assessment of food value in the current patch, which in turn alters patch-leaving dynamics, increasing the exit rates from cephalexin-treated patches as shown in *Figure 3*. The decision of whether to stay or leave a patch of food while foraging has been well studied in *C. elegans*. (*Shtonda and Avery, 2006*; *Bendesky et al., 2011*; *Milward et al., 2011*; *Olofsson, 2014*; *Scheer and Bargmann, 2023*). These studies point to a similar mechanism. The key advance here is the demonstration that the worm's foraging decisions are consistent with formal effort-discounting models under our conditions.

Stay-or-leave decision-making based on assessment of current conditions has been observed in the context of effort discounting in other organisms. One procedure widely used in rodent studies is the *progressive ratio task* (*Salamone et al., 2018*). In this task, animals can either lever press for a preferred food or consume a less preferred food that is freely available nearby. However, the number

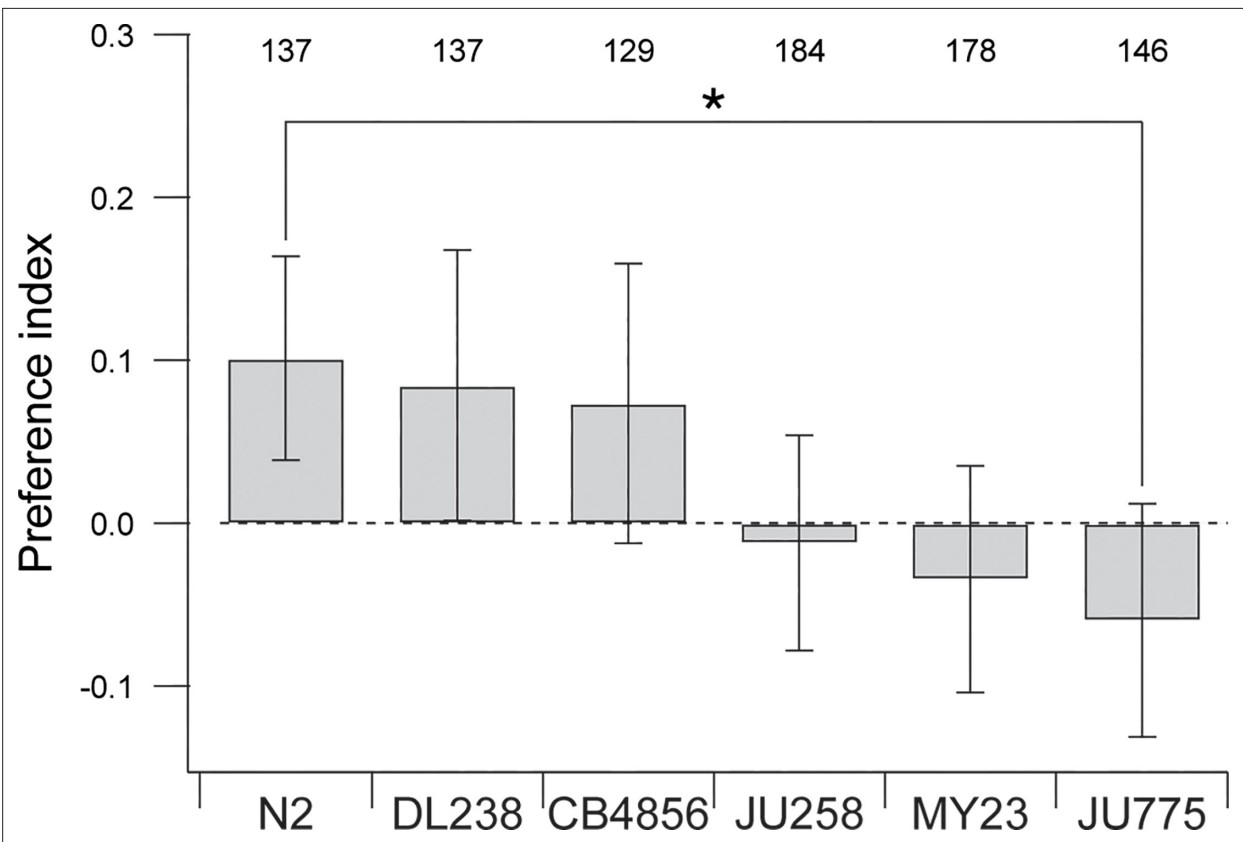

**Figure 5.** Preference for normal versus elongated bacteria in N2 and five wild-isolate strains. Data are mean preferences measured in T-maze assays. N2 and DL238 are significantly different from zero. *Numbers*, sample size; *, $p$ = 1.12E-02. Error bars ± 95% CI.

of lever presses required to obtain preferred food increases as a function of the cumulative number of lever presses until the effort cost of obtaining preferred food becomes too high and the animal switches to a freely available food. This is a foraging assay in the sense that the lever and the freely available food are patches, and the animal decides whether or not to leave the 'lever' patch. Our assay also has parallels in the human effort-discounting literature. In one example (*Bonnelle et al., 2015*), participants are presented with a series of virtual apple trees. They can see how many apples are in the current tree and how much effort (squeezing a handgrip) is required to gather them. Their task is to decide whether or not to gather apples based on the perceived cost and benefit. Here, the trees are equivalent to patches, and the decision to gather apples or not is analogous to a stay–leave decision in patch foraging. Thus, stay-or-leave decision-making may be a conserved solution to effortful foraging tasks.

### Role of dopamine signaling in effort discounting-like behavior

Fully intact dopamine signaling is required for normal effort discounting-like behavior in *C. elegans*. This finding is broadly consistent with the known requirements of dopamine signaling for foraging behavior in *C. elegans*, as our T-maze assay is essentially a foraging assay. *C. elegans* dopamine receptors DOP-1, DOP-2, and DOP-3 are required for normal responses to the presence or absence of food (*Chase and Koelle, 2007*; *Suo et al., 2009*; *Bastien et al., 2024*). Dopamine itself is required for area-restricted search in which worms, suddenly finding themselves outside a food patch, engage in a period of frequent locomotory reorientations in an attempt to relocate the food (*Hills et al., 2004*).

However, the relationship between dopamine signaling and effort discounting-like behavior in *C. elegans* remains an area for future research. For example, although mutations in the dopamine pathway cause a range of locomotor phenotypes (*Chase et al., 2004*; *McDonald et al., 2007*; *Ezak and Ferkey, 2010*) and may affect metabolism (*de Almeida Barros et al., 2014*), only *dop-3* displays a phenotype in our effort-discounting assay, suggesting a specific role of *dop-3* in effort discounting in *C. elegans*. Understanding the precise role of *dop-3* and other components of the dopamine signaling pathway in effort discounting will require more sensitive behavioral assays as well as other technical approaches, including cell-specific rescues. The fact that decreased dopamine signaling in the *dop-3* mutant rendered worms indifferent between the food options suggests that decreased dopamine signaling *increases* tolerance for high-effort food, in contrast to its effects in rodents. This discrepancy may reflect a true difference in the function of dopamine in effort discounting in *C. elegans* and rodents, or methodological differences. In particular, manipulations of dopamine signaling in rodent studies mainly involve acute experiments using pharmacological agents, whereas we studied worms with chronic, lifelong alterations in dopamine signaling.

In summary, our study provides strong evidence that *C. elegans* exhibits effort discounting-like behavior. By demonstrating that worms devalue effortful food like higher organisms, we extend the phylogenetic boundary of this behavior. The application of an economic framework allowed us to quantify this effect and make testable predictions, confirming its robustness. Future research can now exploit the utility of *C. elegans* as a genetically tractable system to explore how environmental factors and genetic variations influence effort-based decision-making. Such an approach could provide deeper insights into the fundamental principles and molecular basis of cost–benefit decisions.

## Materials and methods

**Key resources table**

| Reagent type (species) or resource | Designation | Source or reference | Identifiers | Additional information |
|---|---|---|---|---|
| Strain, strain background (*E. coli*) | OP50 | CGC* | RRID:WB-STRAIN:WBStrain00041969 | Worm maintenance |
| Strain, strain background (*Comamonas* spp.) | DA1877 | CGC | RRID:WB-STRAIN:WBStrain00040995 | Behavioral testing |
| Strain, strain background (*C. elegans*) | N2 | CGC | RRID:WB-STRAIN:WBStrain00000001 | All WT assays |
| Strain, strain background (*C. elegans*) | *cat-2(e1112)* | CB | RRID:WB-STRAIN:WBStrain00004246 | Behavioral testing |
| Strain, strain background (*C. elegans*) | *dat-1(ok157)* | CGC | RRID:WB-STRAIN:WBStrain000333294 | Behavioral testing |
| Strain, strain background (*C. elegans*) | *dop-1(vs101)* | CGC | RRID:WB-STRAIN:WBStrain00026368 | Behavioral testing |

*Continued on next page*

*Continued*

| Reagent type (species) or resource | Designation | Source or reference | Identifiers | Additional information |
|---|---|---|---|---|
| Strain, strain background (*C. elegans*) | *dop-2(vs105)* | CGC | RRID:WB-STRAIN:WBStrain00026373 | Behavioral testing |
| Strain, strain background (*C. elegans*) | *dop-3(vs106)* | CGC | RRID:WB-STRAIN:WBStrain00026374 | Behavioral testing |
| Strain, strain background (*C. elegans*) | Wild isolate | CGC | RRID:WB-STRAIN:WBStrain00005835 | Behavioral testing |
| Strain, strain background (*C. elegans*) | Wild isolate | CGC | RRID:WB-STRAIN:WBStrain00004602 | Behavioral testing |
| Strain, strain background (*C. elegans*) | Wild isolate | CGC | RRID:WB-STRAIN:WBStrain00022850 | Behavioral testing |
| Strain, strain background (*C. elegans*) | Wild isolate | CGC | RRID:WB-STRAIN:WBStrain00027669 | Behavioral testing |
| Strain, strain background (*C. elegans*) | Wild isolate | CGC | RRID:WB-STRAIN:WBStrain00023072 | Behavioral testing |
| Chemical compound, drug | Cephalexin hydrate | Thermo Fisher | J6317206 | Bacteria elongation |
| Other | Oil-Red-O | Sigma-Aldrich | 00625 | Histological stain |
| Other | BacLight Green | Thermo Fisher | B-35000 | Histological stain |
| Software, algorithm | Igor Pro | Wavemetrics | Version 9.05 | Behavioral testing, image analysis |
| Software | R | R Core Team (2024) | Version 4.4.2 (2024-10-31) | Statistical analysis |
| Software, algorithm | WormLab | MBF Bioscience | Version 2024 | Video analysis |

*\*Caenorhabditis Genetics Center.*

## Bacteria culture

Two bacterial species were used: *E. coli* (OP50) is used for worm culture, and *Comamonas* spp. (DA1877) for assays. Bacterial cultures were grown in liquid lysogeny broth (LB) under sterile conditions at 37°C with agitation. For all assays, DA1877 was grown for 1 day, then 'Control' cultures were solubilized with M9 buffer (3 g $KH_2PO_4$, 6 g $Na_2HPO_4$, 5 g NaCl, 1 ml 1 mol/l $MgSO_4$, $H_2O$ to 1 l); 'cephalexin' cultures were supplemented with a solution of M9 and cephalexin hydrate 97% to a concentration of 450 µmol/l and incubated for 2 hr. Cultures then underwent three rounds of concentration by centrifugation, followed by rinsing with 10 ml of M9 buffer before being resuspended to their final concentration. Concentration was defined as OD at 600 nm, as measured with a cell density meter (Laxco, DSM, Bothell, WA, USA). All measurements were performed on samples diluted into the linear range of the instrument (OD 0.1–1.0).

## Worm maintenance and synchronization

All strains were maintained at 20°C on 5 cm plates containing NGM seeded with *E. coli* OP50 bacteria as a food source (*Brenner, 1974*). For synchronization, 20 young adult worms were transferred to fresh 5 cm NGM plates. Adults were removed from the plates after laying eggs for 3 hr. The plates were maintained at 20°C until worms reached the young adult stage.

## Pharyngeal pumping assay

Pharyngeal pumping was measured electrophysiologically using the ScreenChip microfluidic system (InVivo Biosystems, Eugene, OR, USA). Synchronized young adults were preincubated with control or cephalexin-treated bacteria at OD 1 for 20 min before being transferred into the reservoir of a microfluidic device fitted with electrodes. Individual worms were moved one at a time from the reservoir into the recording channel and given 1–2 min to acclimate. Voltage transients associated with pharyngeal pumping were recorded for 5 min; such recordings are called electropharyngeograms. All recordings were performed within 90 min of the worms being loaded into the microfluidic reservoir. Mean pumping frequency was extracted using custom code written in Igor Pro (WaveMetrics, Lake Oswego, OR, USA). This code is included as a Supporting File titled *Source code 1*. The code operates on a set of voltage time courses obtained from a cohort of worms. It identifies the

individual pumping events in each time series and computes the mean pumping frequency across the cohort.

## Food-preference assay

T-mazes (*Levichev et al., 2023*) were constructed by cutting masks from sheets of 2 mm thick ethylene-vinyl acetate foam (Darice Craft Foam, Strongsville, OH, USA) using a fabric-cutting machine (Cricut, South Jordan, UT, USA). Masks were placed on blank 5 cm NGM plates. Maze arms were baited with 4.5 µl of various bacterial suspensions as described in the text. Synchronized young adults were washed by five rounds of centrifugation (30 s, 300 × *g*) followed by aspiration of the supernatant, then deposited by liquid transfer at the starting point of the maze. A transparent plastic disc was placed over the maze to eliminate air currents. Twelve plates were placed on a flatbed scanner (EPSON, model J221B, Los Alamitos, CA, USA) and imaged after an elapsed time of 60 min. Imaged worms were counted manually. The food preference index was computed as $I = (n_N - n_L) / (n_N + n_L)$, where $n_N$ and $n_L$ are the number of worms in the normal and elongated patches, respectively. In some experiments, a paralytic agent (sodium azide, $NaN_3$, 3 µl at 20 mM) was added to each food patch to prevent animals from leaving the patch of food after reaching it. Sodium azide diffuses through the agar over time, and its action is not instantaneous. To account for these effects, all worms within 5 mm of the end of the maze arm, rather than only those on food, were counted when calculating the preference index.

## Patch-leaving assay

Custom 8-well plates (25 mm × 45 mm) were fabricated from acrylic plastic. Wells (diameter: 10 mm; depth: 6 mm) were filled with NGM agar. Bacterial suspension (2 µl, OD 1) was placed at the center of each well, forming a patch 3–4 mm in diameter. Synchronized 1-day-old adult worms were cleaned by transferring them to an unseeded NGM plate for several minutes. One worm was transferred to each well using a pick and placed at the center of the patch. The well plate was immediately placed on a stage 8 cm above a diffuse white-light source and imaged (2 or 5 frames per second, 23.4 µm/pixel) for 15 min using a video camera (AmScope, 1080P, Irvine, CA, USA) fitted with a telephoto lens (Nikon USA, AF Micro Nikkor 60 mm, Melville, NY, USA). The distance between the lens and the stage was 50 cm. Worm tracks and locomotion parameters were extracted using WormLab software (MBF Bioscience, Williston, VT, USA) and analyzed using custom routines in Igor Pro (WaveMetrics, Lake Oswego, OR, USA). To define the worm's position with respect to the patch, its margin was traced manually.

## Bacterial composition analysis

Bacteria were grown under sterile conditions at 37°C with agitation in 500 ml of LB overnight, pelleted (5000 rpm, 3 min), washed with water, and re-pelleted. Pellets were stored at –80°C prior to analysis. Bacterial stocks were weighed and then lysed with CelLytic B Plus kit (Sigma-Aldrich, CB0050, Saint Louis, MO, USA) on the day of the assay.

### Protein

Protein was assayed with the Pierce BCA Protein Assay kit (Thermo Fisher Scientific, A55864, Waltham, MA, USA). Readings were taken with a spectrophotometer (NanoDrop, Thermo Fisher Scientific, ND-2000, Waltham, MA, USA) using the bicinchoninic acid assay setting.

### Carbohydrate

Carbohydrate was measured with the Abbexa Total Carbohydrate Assay Kit (Abbexa, abx298986, Cambridge, UK). Spectrophotometer readings were made at OD 540 nm.

### Lipid

Lipid was quantified using the method of *Aggeler et al., 1987*. We prepared 100 µl of cell lysate and added: 500 µl methanol, 250 µl dichloromethane, and 100 µl distilled $H_2O$. The mixture was vortexed before adding an additional 375 µl of distilled water. The solution was vortexed again to form an emulsion and then centrifuged at 4800 × *g* for 10 min. The bottom layer of solution was transferred to

a pre-weighed tube and then weighed again after the lipid layer dried. The difference in weight was used to measure the quantity of lipid per 100 µl of cell lysate.

## Bacteria visualization

Control and cephalexin-treated bacteria were prepared as described above. BacLight Green dye dissolved in DMSO was added to the bacteria at a final concentration of 0.1 µM. Following a 15-min incubation, bacteria were mounted between a microscope slide and a coverslip, then imaged on a spinning disk confocal microscope (Nikon Eclipse Ti-2E with Yokogawa CSU-W1; objective CFI60 Plan Fluor 40x Oil Immersion, N.A. 1.3; Melville, NY, USA).

## Fat storage analysis

Fat storage was assessed using the procedures of *Soukas et al., 2009* and *Stuhr and Curran, 2020*. A 0.5% isopropanol solution of Oil-Red-O (Sigma-Aldrich, Oil-Red-O powder 00625, Saint Louis, MO, USA) was prepared several days before the assay. Bacterial cultures (Control and cephalexin) were prepared as described above and adjusted to OD 3 before being loaded onto fresh NGM plates. Seventeen young adult worms were placed on these plates and allowed to lay eggs for 5 hr before being removed. The plates were incubated at 20°C for 2 days. L4 larvae were harvested and washed in PBS + 0.01% Triton X-100. Worms were fixed with a 60% isopropanol solution and incubated with the Oil-Red-O solution for 2 hr on a rotating plate. Worms were washed twice with PBS + Triton 0.01% solution, first for 30 min, then for 1 hr, on a rotating plate. Stained worms were imaged on a microscope slide at 32×, and staining intensity was quantified using custom routines in Igor Pro (WaveMetrics, Lake Oswego, OR, USA).

## Statistical analysis

### General

Statistical details for each test (figure panel, statistical test, effect or comparison tested, definition of units of replication, number of replicates, statistic value, degrees of freedom, p-value, and effect size) are compiled in *Table 1*. These details are cited in the text using the notation '*Table 1*, row *n*', where *n* is the row number in the table. For all comparisons, the centers of distributions were their arithmetic mean or median. For dispersion measures, we used 95% confidence intervals. The following statistical tests were employed according to the experimental design: We used *t*-tests to determine whether the mean differed from zero. To assess differences in pairs of means, we used *t*-tests and, when the data were not normally distributed (according to the Shapiro–Wilk test), the Mann–Whitney *U* test. For comparisons of three or more means, we used one- or two-way ANOVA with post hoc tests.

### State probabilities

State probabilities are an instance of *compositional data*, meaning data that sum to 1 (or 100%), which requires special treatment. For analysis, we subjected these data to the isometric log ratio transform (*Aitchison, 1986*; *Egozcue et al., 2003*) using the 'compositions' package in R (*van den Boogaart and Tolosana-Delgado, 2008*). Following transformation, *t*-tests were used to infer statistical significance (*Figure 3—figure supplement 1*).

### Box plots

The box represents the first and third quartiles of the data, with a horizontal line representing the median. The whiskers extend to the maximum and minimum values following the removal of extreme outliers. These were defined as points with values below $Q1 - 3 \times IQR$ or above $Q1 + 3 \times IQR$, where $Q1$ is the first quartile, $Q3$ the third quartile, and $IQR$ is the interquartile range $Q3 - Q1$. A diamond symbol represents the mean.

### Number of replicates

In the case of *assay plates*, a replicate is a unique cohort of worms on an assay plate, such as a T-maze. In the case of *worms*, a replicate is a unique animal tested once. Nutrient replicates (*Figure 1C–E*) were biological or technical, as indicated in *Table 1*. The minimum sample size for the T-maze assays was based on pilot experiments that demonstrated the ability to detect moderate to small effects

with 10–30 replicates per experimental condition. Previously published electropharyngeogram data (*Katzen et al., 2023*) showed that mutants or treatments could be distinguished with approximately 20 replicates; however, to ensure the detection of small effect sizes across experimental conditions, larger sample sizes were used.

## Effect size

For *t*-tests, we used Cohen's *d*. For Mann–Whitney *U* tests, we used the rank-biserial correlation with $r = 2U/\left(n_1 n_2\right) - 1$, where $U$ is the Mann–Whitney statistic and $n_1$ and $n_2$ are group sizes. For one-way ANOVA, we used eta squared with $\eta^2 = SS_\text{between}/SS_\text{total}$ where $SS_\text{between}$ is the sum of squares between groups and $SS_\text{total}$ is the total sum of squares. For two-way ANOVA, we used $\eta^2 = SS_\text{effect}/SS_\text{total}$ where $SS_\text{effect}$ is the sum of squares for a factor or interaction. For MANOVA, we used partial eta squared with $\eta_p^2 = SS_\text{effect}/\left(SS_\text{effect} + SS_\text{error}\right)$ where $SS_\text{error}$ is sum of squares for error.

## Acknowledgements

This research was supported by NIH grants DA053817 and GM152169. Some strains were provided by the CGC, which is funded by NIH Office of Research Infrastructure Programs (P40 OD010440).

## Additional information

### Competing interests

Shawn R Lockery: has a competing financial interest in InVivo Biosystems, Inc, which produces the ScreenChip system utilized in the pharyngeal pumping assay. The author has no other competing interests to declare. The other authors declare that no competing interests exist.

### Funding

| Funder | Grant reference number | Author |
|---|---|---|
| National Institute on Drug Abuse | DA053817 | Shawn R Lockery |
| National Institute of General Medical Sciences | GM152169 | Shawn R Lockery |

The funders had no role in study design, data collection, and interpretation, or the decision to submit the work for publication.

### Author contributions

Jonathan RM Millet, Conceptualization, Data curation, Investigation, Methodology, Writing – original draft; Serge Faumont, Data curation, Investigation; Aaron B Schatz, Amanda M White, Kathy D Chicas-Cruz, Investigation; Shawn R Lockery, Conceptualization, Formal analysis, Supervision, Funding acquisition, Writing – original draft, Project administration, Writing – review and editing

### Author ORCIDs

Shawn R Lockery ⓘ https://orcid.org/0000-0001-8535-7989

Reviewer #1 (Public review): https://doi.org/10.7554/eLife.106792.3.sa1
Reviewer #2 (Public review): https://doi.org/10.7554/eLife.106792.3.sa2
Reviewer #3 (Public review): https://doi.org/10.7554/eLife.106792.3.sa3
Author response https://doi.org/10.7554/eLife.106792.3.sa4

## Additional files

### Supplementary files

MDAR checklist

Source code 1. Electropharyngeogram analysis routines.

Source data 1. Raw data for *Figure 1A, B, F2*, *Figure 2B, C*, *Figure 3C, D*, *Figure 4* and *Figure 5*.

### Data availability

All data generated in this study are included in the manuscript and the supporting file titled *Source data 1*. This Microsoft Excel file contains the raw data for: Figure 1A, B, Figure 1F2, Figure 2B,C, Figure 3C,D, Figure 4, and Figure 5. Computer code written to analyze the data in Figure 1A, written in Igor Pro, is included in the supporting file titled *Source code 1*. This code identifies pharyngeal pumps in electrophysiological data and computes mean pump frequency.

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
