## [Editor Report · eLife Assessment]

This **important** work develops the *C. elegans* as a model organism for studying effort-based discounting by asking the worms to choose between patches of easy and hard to digest bacteria. The authors provide **convincing** evidence that the nematodes are effort discounting. They also provide **solid** evidence of involvement of dopamine in the food preference and that the finding is not restricted to lab-acclimated strains.

---

## [Referee Report · Reviewer #1 (Public review)]

Summary:

Millet et al. show that *C. elegans* systematically prefers easy-to-eat bacteria but will switch its choice when harder-to-eat bacteria are offered at higher densities, producing indifference points that fit standard economic discounting models. Detailed kinetic analysis reveals that this bias arises from unchanged patch-entry rates but significantly elevated exit rates on effortful food, and dop-3 mutants lose the preference altogether, implicating dopamine in effort sensitivity. These findings extend effort-discounting behavior to a simple nematode, pushing the phylogenetic boundary of economic cost-benefit decision-making.

Strengths:

Extends the well-characterized concept of effort discounting into *C. elegans*, setting a new phylogenetic boundary and opening invertebrate genetics to economic-behavior studies.

Elegant use of cephalexin-elongated bacteria to manipulate "effort" without altering nutritional or olfactory cues, yielding clear preference reversals and reproducible indifference points.

Application of standard discounting models to predict novel indifference points is both rigorous and quantitatively satisfying, reinforcing the interpretation of worm behavior in economic terms.

The three-state patch-model cleanly separates entry and exit dynamics, showing that increased leaving rates-rather than altered re-entry-drive choice biases.

Demonstrates that _dop-3_ mutants lose normal effort discounting, firmly tying monoaminergic signaling to this behavior and paralleling vertebrate findings.

Demonstration of discounting in wild strain (solid evidence).

Weaknesses:

Only _dop-3_ shows an effect, whereas _cat-2_/_dat-1_ do not, leaving the broader role of dopamine synthesis and reuptake ambiguous.

With only five wild isolates tested, and only one clearly showing clear evidence of preference for the easy to eat bacteria, it's hard to conclude that effort discounting isn't a lab-strain artifact or how broadly it varies in natural populations.

---

## [Referee Report · Reviewer #2 (Public review)]

Summary:

Here Millet et al. adapted a t-maze paradigm for use in *C. elegans* to understand whether nematodes exhibit effort discounting behaviors comparable to other species. *C. elegans* worms were reliably sensitive to how effortful the food was to consume, allowing for the application of standard economic models of decision-making to be applied to their behavior. The authors then demonstrated the necessity of dopamine signaling for this behavior, identifying dop-3 mutants in particular as insensitive to effort. Together, this work establishes a new model system for the study of discounting behavior in cost-benefit decision-making.

Strengths:

The question is well-motivated and the approach taken here is novel; it is uncommon for worms to undergo such behavioural procedures (although this lab has previously been integral to pushing the extent of the complexity of behaviours studied in *C. elegans*). The authors are careful in their approach to altering and testing the properties of the elongated bacteria. Similarly, they go to some effort to understand what exactly is driving behavioural choices in this context, both through application of simple standard models of effort discounting and a kinetic analysis of patch leaving. The comparisons to various dopamine mutants further extends the translational potential of their findings. I also appreciate the comparison to natural isolate strains as the question of whether this behaviour may be driven by some sort of strain-specific adaptation to the environment is not regularly addressed in mammalian counterparts to this work.

Weaknesses:

The authors have now addressed concerns about whether the mechanisms underlying the choice behavior here are generalizable to other organisms. Specifically, their work speaks to foraging-inspired effort discounting paradigms in rodents and humans in which the decision is whether to stay or leave a given resource, rather than to simultaneous decision-making across two options in a T-maze.

The dopamine results are interesting but still difficult to interpret. As the authors discuss, the lack of an effect in the cat-2 and dat-1 mutants is surprising given the effect in the dop-3 mutants. Understanding what exactly the role of dop-3 is here therefore requires further study.

---

## [Referee Report · Reviewer #3 (Public review)]

Summary:

The authors establish a behavioral task to explore effort discounting in *C. elegans*. By using bacterial food that takes longer to consume, the authors show that for equivalent effort, as measured by pumping rate, animals obtain less food, as measured by fat deposition.

The authors formalize the task by applying a neuroeconomic decision making model that includes, value, effort, and discounting. They use this to estimate the discounting *C. elegans* apply based on ingestion effort by using a population level 2-choice T-maze.

They then analyze the behavioral dynamics of individual animals transitioning between on-food and off-food states. Harder to ingest bacteria led to increased food patch leaving.

Finally, they examined a set of mutants defective in different aspects of dopamine signaling, as dopamine plays a key role in discounting in vertebrates and regulates certain aspects of *C. elegans* foraging.

In their response to the first set of reviews, the authors take care to ensure their task is analogous to at least some of those used in mammals and make changes to the text to better clarify some of their conclusions. My view is the same--that this is an interesting paper for methodological and scientific reasons that brings an important theoretical framework to bear on *C. elegans* foraging behavior. While I think the mutant results are somewhat unsatisfying, this is not the principal contribution of the work.

Strengths:

The behavioral experiments and neuroeconomic analysis framework are compelling and interesting and make a significant contribution to the field. While these foraging behaviors have been extensively studied, few include clearly articulated theoretical models to be tested.

Demonstrating that *C. elegans* effort discounting fits model predictions and has stable indifference points is important for establishing these tasks as a model for decision making.

Weaknesses:

The dopamine experiments are harder to interpret. The authors point out the perplexing lack of an effect of dat-1 and cat-2. dop-3 leads to general indifference. I am not sure this is the expected result if the argument is a parallel functional role to discounting in vertebrates. dop-3 causes a range of locomotor phenotypes and may affect feeding (reduced fat storage), and thus there may be a general defect in the ability to perform the task rather than anything specific to discounting.

That said, some of the other DA mutants also have locomotor defects and do not differ from N2. But there is no clear result here-my concern is that global mutants in such a critical pathway exhibit such pleiotropy that it's difficult to conclude there is a clear and specific role for DA in effort discounting. This would require more targeted or cell-specific approaches. The authors state these experiments are outside the scope of the current study, and that at minimum their results implicate dopamine signaling in some form. I tend to agree but still think locomotion defects of DA mutants complicate this question.

Meanwhile, there are other pathways known to affect responses to food and patch leaving decisions-5HT, PDF, tyramine, etc. in their response the authors state they focus on dopamine because of its role in discounting behavior in mammals.

---

## [Author Response]

The following is the authors’ response to the original reviews.

**Reviewer #1(Public Reviews):**
Summary:Here, Millet et al. consider whether the nematode *C. elegans* 'discounts' the value of reward due to effort in a manner similar to that shown in other species, including rodents and humans. They designed a T-maze effort choice paradigm inspired by previous literature, but manipulated how effortful the food is to consume.*C. elegans* worms were sensitive to this novel manipulation, exhibiting effort-discountinglike behaviour that could be shaped by varying the density of food at each alternative in order to calculate an indifference point. This discounting-like behaviour was related to worms' rates of patch leaving, which differed between the low and high effort patches in isolation. The authors also found a potential relationship to dopamine signalling, and also that this discounting behaviour was not specific to lab-based strains of *C. elegans* .Strengths:The question is well-motivated, and the approach taken here is novel. The authors are careful in their approach to altering and testing the properties of the effortful, elongated bacteria. Similarly, they go to some effort to understand what exactly is driving behavioural choices in this context, both through the application of simple standard models of effort discounting and a kinetic analysis of patch leaving. The comparisons to various dopamine mutants further extend the translational potential of their findings. I also appreciate the comparison to natural isolate strains, as the question of whether this behaviour may be driven by some sort of strain-specific adaptation to the environment is not regularly addressed in mammalian counterparts. The manuscript is well-written, and the figures are clear and comprehensible.Weaknesses:Discounting is typically defined as the alteration of a subjective value by effort (or time, risk, etc.), which is then used to guide future decision-making. By adapting the standard t-maze task for *C. elegans* as a patch-leaving paradigm, the authors observe behaviour strongly consistent with discounting models, but that is likely driven by a different process, in particular by an online estimate of the type of food in the current patch, which then influences patch-leaving dynamics (Figure 3). This is fundamentally different from decision-making strategies relating to effort that have been described in the rodent and human literatures.

We agree that in our study worms are likely making an on-line estimate of food quality in the current patch, but we wish to point out that rodents and humans also use on-line estimates in some significant effort-discounting paradigms. With respect to rodents, we call attention to effort discounting studies involving the widely used progressive ratio task (references in Discussion). In this task, animals can either lever-press for a preferred food or consume a less preferred food that is freely available nearby. However, the number of lever presses required to obtain preferred food increases as a function of the cumulative number of lever presses until the effort-cost of obtaining preferred food becomes too high and the animal switches to a freely available food. In essence, the lever and the freely available food are patches and the animal decides whether or not to leave the “lever” patch. It seems inescapable that the progressive ratio task involves an on-line assessment of the cost/benefit relationship associated with lever pressing. With respect to humans, one highly cited study (reference in Discussion) presented participants with a series of virtual apple trees. They could see how many apples are in the current tree and how much effort (squeezing a handgrip) is required to gather them. Their task was to decide whether or not to gather apples from that tree based on the perceived cost and benefit. Thus, on-line estimation is a common strategy used by animals and humans as shown in the effort discounting literature. We now make this point in the Discussion section titled A model of effort-discounting like behavior.

Similarly, the calculation of indifference points at the group instead of at the individual level also suggests a different underlying process and limits the translational potential of their findings. The authors do not discuss the implications of these differences or why they chose not to attempt a more analogous trial-based experiment.

It is not clear to us why changing the read-out –– from the individual level to the population level –– necessarily suggests that a different biological mechanism is at work. In our view, there is one mechanism and it can be seen from different perspectives (e.g., individual vs population). Furthermore, the analogous trial-based experiment, as we understand it, would be to record behavior one worm at a time in the T-maze. This design is not practical because it entails recording a large number of single worms in the T-maze for 60 min each.

In the case of both the dopamine and natural isolate experiments, the data are very noisy despite large (relative to other *C. elegans* experiments) sample sizes. In the dopamine experiment, disruption of dop1, dop-2, and cat-2 had no statistically significant effect. There do not appear to be any corrections for multiple comparisons, and the single significant comparison, for dop-3, had a small effect size.

An ANOVA followed by a Dunnett test was used to test differences between groups in Fig. 4 and 5. The Dunnett test is a multiple comparison test comparing experimental groups to a single control group. It is used to minimize type I error while maintaining statistical power and does not require further correction for multiple comparisons. We have clarified the use of the Dunnett test in the statistical table. The effect size for dop-3 is 0.5 (Cohen’s d), which is typically interpreted as a medium, not small, effect size.(e.g. Cohen, Psychological Bulletin, 1992, Vol. 112. No. 1,155-159).

More detailed behavioural analyses on both these and the wild isolate strains, for example by applying their kinetic analysis, would likely give greater insight as to what is driving these inconsistent effects.

More detailed behavioral analysis could reveal why we observe a difference in effort discounting in some strains and not others. However, it is not obvious what type of behavioral analysis would be needed to differentiate between pleiotropic effects of the mutations/natural isolates and more specific effects on effort discounting. A simple kinetic analysis in particular may not be enough to reveal relevant differences between mutants/natural isolates. For this reason, we think that such experiments may be better suited for future follow up studies.

**Reviewer #2 (Public Reviews)**
Summary:Millet et al. show that *C. elegans* systematically prefers easy-to-eat bacteria but will switch its choice when harder-to-eat bacteria are offered at higher densities, producing indifference points that fit standard economic discounting models. Detailed kinetic analysis reveals that this bias arises from unchanged patch-entry rates but significantly elevated exit rates on effortful food, and dop-3 mutants lose the preference altogether, implicating dopamine in effort sensitivity. These findings extend effortdiscounting behavior to a simple nematode, pushing the phylogenetic boundary of economic costbenefit decision-making.Strengths:(1) Extends the well-characterized concept of effort discounting into *C. elegans* , setting a new phylogenetic boundary and opening invertebrate genetics to economic-behavior studies.(2) Elegant use of cephalexin-elongated bacteria to manipulate "effort" without altering nutritional or olfactory cues, yielding clear preference reversals and reproducible indifference points.(3) Application of standard discounting models to predict novel indifference points is both rigorous and quantitatively satisfying, reinforcing the interpretation of worm behavior in economic terms.(4) The three-state patch-model cleanly separates entry and exit dynamics, showing that increased leaving rates-rather than altered re-entry-drive choice biases.(5) Investigates the role of dopamine in this behavior to try to establish shared mechanisms with vertebrates.(6) Demonstration of discounting in wild strain (solid evidence).Weaknesses:(1) The kinetic model omits rich trajectory details-such as turning angles or hazard functions-that could distinguish a bona fide roaming transition from other exit behaviors.

The overarching goal of present paper was to develop a simple model for effort discounting in a small, genetically tractable organism. Accordingly, we focused on quantitative assays that are easy to implement and analyze. The patch-leaving assay and its associated kinetic analysis are one such assay. To keep things simple in this assay, we counted the number of transitions between the three states shown in Fig. 3A. We chose not to analyze the data in terms of turning angles or hazard functions because the metrics we developed seemed sufficient. Finally, we note that there are new modeling data showing that the presumptive transitions into the roaming state can be explained in terms of a one-state stochastic model in which there is no discrete roaming state (Elife. 2025 Jul 30;14:RP104972. doi: 10.7554/eLife.104972.PMID: 40736321).

(2) Only dop-3 shows an effect, and the statistical validity of this result is questionable. It is not clear if the authors corrected for multiple comparisons, and the effect size is quite small and noisy, given the large number of worms tested. Other mutants do not show effects. Given these two concerns, the role of dopamine in *C. elegans* effort discounting was unconvincing.

An ANOVA followed by a Dunnett test was used to test statistical significance in figures 4 and 5 (see above for a discussion of these tests). We believe this approach is rigorous, and the use of these tests is statistically valid. We note that the effect size for this comparison was medium.

(3) With only five wild isolates tested (and variable data quality), it's hard to conclude that effort discounting isn't a lab-strain artifact or how broadly it varies in natural populations.

The fact that four of the five natural isolates tested display levels of effort discounting similar to N2 (only one natural isolate does not display effort discounting) argues against effort discounting being a laboratory adaption. We have nevertheless weakened the claim regarding natural isolates. We now say effort discounting-like behavior may not be an adaptation to the laboratory environment.

(4) Detailed analysis of behavior beyond preference indices would strengthen the dopamine link and the claim of effort discounting in wild strains.

Going beyond preference in the behavioral analysis might or might not reveal new phenotypes that strengthen the link with dopamine. At present, however, we think such experiments are beyond the scope of the paper.

(5) A few mechanistic statements (e.g., tying satiety exclusively to nutrient signals) would benefit from explicit citations or brief clarifications for non-worm specialists.

We are unable to identify a mechanistic statement tying satiety to nutrient signals in our manuscript.

**Reviewer #3 (Public Reviews)**
Summary:The authors establish a behavioral task to explore effort discounting in *C. elegans*. By using bacterial food that takes longer to consume, the authors show that, for equivalent effort, as measured by pumping rate, they obtain less food, as measured by fat deposition. The authors formalize the task by applying a formal neuroeconomic decision-making model that includes value, effort, and discounting. They use this to estimate the discounting that *C. elegans* applies based on ingestion effort by using a population-level 2-choice T-maze. They then analyze the behavioral dynamics of individual animals transitioning between on-food and off-food states. Harder to ingest bacteria led to increased food patch leaving. Finally, they examined a set of mutants defective in different aspects of dopamine signaling, as dopamine plays a key role in discounting in vertebrates and regulates certain aspects of *C. elegans* foraging.Strengths:The behavioral experiments and neuroeconomic analysis framework are compelling, interesting, and make a significant contribution to the field. While these foraging behaviors have been extensively studied, few include clearly articulated theoretical models to be tested.Demonstrating that *C. elegans* effort discounting fits model predictions and has stable indifference points is important for establishing these tasks as a model for decision making.Weaknesses:The dopamine experiments are harder to interpret. The authors point out the perplexing lack of an effect of dat-1 and cat-2. dop-3 leads to general indifference. I am not sure this is the expected result if the argument is a parallel functional role to discounting in vertebrates. dop-3 causes a range of locomotor phenotypes and may affect feeding (reduced fat storage), and thus, there may be a general defect in the ability to perform the task rather than anything specific to discounting.That said, some of the other DA mutants also have locomotor defects and do not differ from N2. But there is no clear result here - my concern is that global mutants in such a critical pathway exhibit such pleiotropy that it's difficult to conclude there is a clear and specific role for DA in effort discounting. This would require more targeted or cell-specific approaches.

We agree with the reviewer that the results of the dopamine experiments are puzzling and getting a better understanding of the role of dopamine in effort-discounting will require more sensitive assays and different experimental approaches (e.g. cell-specific rescues). However, as mentioned by the reviewer, all the mutations tested have some pleiotropic effects, yet only dop-3 displays a defect in effort discounting. This, in our opinion, points to a specific role of dop-3 in effort-discounting in *C. elegans*. This point is now made in the Discussion in the section titled Role of dopamine signaling in effort discountinglike behavior.

Meanwhile, there are other pathways known to affect responses to food and patch leaving decisions: serotonin, pigment-dispersing factor, tyramine, etc. The paper would have benefited from a clarification about why these were not considered as promising candidates to test (in addition to or instead of dopamine).

We focused on DA because of its well-established effect on effort discounting in rodents.

Testing other pathways is a goal for future research.

**Reviewer #1 (Recommendations for the authors):**
The current results are more a reframing of data gathered from a patch-leaving paradigm, but described in the form of economic choice modelling in which discounting is one possible explanation. One more parsimonious explanation that worms estimate in real-time some rate of reward and leave the patch at some threshold, consistent with canonical foraging models, previous experiments in *C. elegans*, and the authors' own data (Figure 3). Therefore, I am wary about some of the claims made in this manuscript, such as 'decision-making strategies based on effort-cost trade-offs are evolutionarily conserved'.

These points are now addressed in the Discussion in a revised section titled A model of effortdiscounting like behavior. (i) We now call attention to the fact that our T-maze assay is a patch-leaving foraging paradigm. (ii) We now propose a revised model in which “worms make an on-line assessment of food value in the current patch which in turn alters patch-leaving dynamics, increasing the exit rates from cephalexin-treated patches as shown in Figure 3.” (iii) We now provide evidence from the rodent and human literature that the strategy of on-line assessment of reward value may be evolutionarily conserved in the case of a class of effort discounting tasks whose solution requires on-line assessments.

If the reason the authors chose to do a patch-leaving style task rather than a traditional t-maze is because *C. elegans* is unable to retain the sort of information necessary to make such simultaneous decisions - e.g., if pre-training on the two options isn't possible - then this in itself suggests that mechanisms underlying these decisions in worms and mammals are unlikely to be the same. I mention this because I would like to suggest to the authors an alternative interpretation: that patch foraging is actually 'the' canonical computation that translates across species. This would, in fact, be nicely consistent with some other recent modelling work in humans, e.g., https://www.biorxiv.org/content/10.1101/2025.05.06.652482v1.

Please see the previous response.

**Reviewer #2 (Recommendations for the authors)**:Can you provide a picture of the regular and CEPH bacteria?

Done (see Figure 1––figure supplement 1).

**Reviewer #3 (Recommendations for the authors):**
I would recommend testing representative mutants in other pathways in the choice task. If possible, more targeted experiments with dop-3, including either cell-specific KOs or rescues, would very much strengthen this aspect of the paper.

While valuable, these experiments are out of scope for the present study.